# Current Approach to the Pathogenesis of Diabetic Cataracts

**DOI:** 10.3390/ijms24076317

**Published:** 2023-03-28

**Authors:** Małgorzata Mrugacz, Magdalena Pony-Uram, Anna Bryl, Katarzyna Zorena

**Affiliations:** 1Department of Ophthalmology and Eye Rehabilitation, Medical University of Bialystok, Waszyngtona 17, 15-274 Bialystok, Poland; 2Department of Ophthalmology, Subcarpathian Hospital in Krosno, Korczynska 57, 38-400 Krosno, Poland; 3Department of Immunobiology and Environment Microbiology, Institute of Maritime and Tropical Medicine, Medical University of Gdansk, Dębinki 7, 80-210 Gdańsk, Poland

**Keywords:** cataract, diabetes, sorbitol pathway, oxidative stress, autophagy lysosomal pathway

## Abstract

Cataracts remain the first or second leading cause of blindness in all world regions. In the diabetic population, cataracts not only have a 3–5 times higher incidence than in the healthy population but also affect people at a younger age. In patients with type 1 diabetes, cataracts occur on average 20 years earlier than in the non-diabetic population. In addition, the risk of developing cataracts increases with the duration of diabetes and poor metabolic control. A better understanding of the mechanisms leading to the formation of diabetic cataracts enables more effective treatment and a holistic approach to the patient.

## 1. Introduction

Worldwide, over 285 million people suffer from diabetes. According to the International Diabetes Federation, this number is expected to increase to 439 million by 2030 [1]. Diabetic eye diseases comprise a number of pathological changes within the eye organ, which include, among others: cataracts, retinopathy, diabetic macular edema, dysfunctions of the tear film, and changes in corneal morphology.

Due to the steadily increasing number of people with type 1 and type 2 diabetes, the incidence of diabetic cataracts has increased accordingly. Changes in the lens of diabetic patients resemble age-related modifications in elderly patients. The only difference is that these changes usually manifest themselves at a younger age. 

A cataract is a form of opacification that develops in the translucent lens of the eye, ranging from partial to complete. This condition reduces the translucency of the lenses, which results in reduced visual acuity for the patient, while making it difficult for the ophthalmologist to observe the posterior section of the eye and even making treatment impossible, as in the case of the photocoagulation of the retina. The results of clinical research have shown that cataract development occurs more frequently and at an earlier age in diabetic compared to non-diabetic patients [2,3]. This fact is confirmed by a study of Memon et al., according to which, under 40 years, 33.3% of diabetics, as compared to 16.4% of non-diabetics, develop cataracts; the percentages were 41% and 14.5% in the 40–59 years age group, respectively. The same trend was followed in the age group > 60 years, where 16.4% of non-diabetics develop cataracts compared to 47% of diabetics [3]. Similar results were obtained by Becker et al. in a retrospective observational study involving 56,510 diabetes patients [4]. The estimated incidence rates of cataracts were 20.4 (95% CI, 19.8–20.9) per 1000 person-years (py) in patients with diabetes and 10.8 (95% CI, 10.5–11.2) per 1000 py in the general population. 

There are three types of cataracts: cortical (CC), nuclear (NS), and posterior subcapsular (PSC). Each may occur independently, or, more commonly, they co-occur with each other. Many studies have examined the likelihood of the occurrence of different types of cataracts in diabetic patients. Olafsdottir et al. showed that the prevalences of significant cortical, posterior subcapsular, and nuclear cataracts were 65.5%, 42.5%, and 48.0%, respectively, in the type 2 diabetes population [5]. These results are consistent with most of the previous studies [6,7,8,9,10]. Two studies reported contrasting findings, that is, NS was identified as the predominant subtype [3,11]. Cortical cataract occurrence is often independent of sugar level fluctuations (Figure 1A) [5,12], while posterior subcapsular cataracts are usually associated with poor glycemic control (Figure 1B,C) [5,13]. 

The association between T1DM and cataracts is well documented in the literature [14], with the risk factors being the duration of diabetic symptoms before diagnosis, diabetic ketoacidosis, poor metabolic control, elevated HbA1c, possible genetic factors, and treatment with glucocorticoids. Cataracts affect from 0.7% to 3.4% of type 1 diabetes patients from different populations, and the average 10-year cumulative incidence of cataract surgery, according to a study, is 8.5% [15,16,17]. Regarding the morphology of early diabetic cataracts, Wilson et al. reported multiple morphologies, including posterior subcapsular, lamellar, cortical, snowflake, and milky white types of early diabetic cataracts [17], while most other authors discriminate fewer types with posterior subcapsular cataract described as the most common type of diabetic cataract in childhood [18,19].

The impact of diabetes on the lens also presents itself in accommodation disorders, leading to a decrease in its amplitude and the occurrence of presbyopia at a younger age in comparison to those without diabetes.

Numerous studies have indicated that refractive errors may be associated with glycemic control in diabetic patients [20,21]. Sudden refractive changes toward myopia may indicate undiagnosed or poorly controlled diabetes. On the other hand, in intensive therapy lowering blood glucose levels, a significant number of patients have a tendency toward hyperopia [22]. This effect is particularly pronounced in patients receiving better control post hyperglycemia, who may develop a change in hyperopia ranging from 0.5 diopters (D) to 3.75 D for 1–2 weeks after the beginning of their treatment [23]. Changes in refraction observed during periods of unstable glucose levels are believed to be related to both morphological and functional changes in the lens [24]. In addition, changes in corneal topographic parameters during glucose fluctuations are a potential source of refractive errors after cataract surgery [25]. An increase in central corneal thickness has been reported in the eyes of diabetic patients, which may alter corneal power [26]. An extended duration of diabetes (>15 years) was found to be a significant factor for astigmatism (aOR, 1.62; 95% CI, 1.15–2.27) and anisometropia (aOR, 1.87; 95% CI, 1.29–2.71) [27].

Currently, the main method of treating diabetic cataracts is the surgical removal of the cataract with intraocular lens implantation. Approximately 20% of all cataract surgeries are performed in diabetic patients [28]. Even though cataract surgery is highly effective and implantation of an artificial lens eliminates the possibility of refractive changes, unfortunately, diabetic patients are at a higher risk of intraoperative and postoperative complications compared to non-diabetic patients [29,30]. It has been shown that phacoemulsification in diabetic patients can lead to a relatively rapid progression of diabetic retinopathy, accelerate vitreous hemorrhage [31], and increase the thickness of the central retina [32]. Even uncomplicated cataract surgery leads to an increase in the concentration of inflammatory cytokines, vascular endothelial growth factor (VEGF), hepatocyte growth factor, interleukin-1 (IL-1), and pigment-epithelium-derived factor [33]. It is not surprising that diabetic patients show an increased risk of developing endophthalmitis [34]. Therefore, both the underlying disease itself and the necessity to undergo surgery constitute a significant health and economic burden for the patients, especially in developing countries where diabetes treatment is insufficient and cataract surgery is often unavailable [35]. Therefore, explaining the pathomechanisms that delay or prevent cataract development in diabetic patients remains challenging.

### 1.1. Diabetic Cataract Pathogenesis

The pathogenesis of diabetic cataracts is still not fully understood and remains the subject of many studies. It is worth noting that the pathomechanism of diabetic changes in the lens is different from diabetic complications of tissues of the retina, the peripheral nerves, the kidneys, or the heart. This is because the lens is not vascularized, and is supplied with nutrients and oxygen via the aqueous humor [36]. This type of construction results in obtaining energy mainly anaerobically. In turn, energy production in mitochondria is not as essential as in vascular tissues and is limited to the mitochondria of the single-layer lens epithelium [37]. The key role of hyperglycemia in the formation of changes in the lens of diabetic patients is also important, whereas factors such as hypertension or lipid metabolism disorders, which contribute to the formation of diabetic changes in the vascularized tissues of the eye, are not significant in the pathogenesis of cataracts. 

As clinical practice demonstrates, diabetic complications do not develop in a certain percentage of patients, while in others, even with adequate metabolic control, the progression of complications is surprisingly fast. These observations indicate that there are several risk factors that may affect the likelihood of developing diabetic cataracts [38]. 

In this review, we will develop the hypothesis that diabetic cataract formation can be caused by multifactorial reactions and pathways interacting with each other.

### 1.2. The Sorbitol Pathway

Metabolism in the lens works to maintain its translucency. Glucose metabolism is the main source of energy in the lens. Glucose enters the lens from the aqueous humor both directly by means of simple diffusion and indirectly by facilitated diffusion. The majority of the glucose entering the lens is phosphorylated and converted into glucose-6-phosphate (G6P) by hexokinase. The resulting G6P undergoes further transformation in one of two ways, of which anaerobic glycolysis constitutes approximately 80%. The glucose that is not phosphorylated to G6P enters the sorbitol cycle, where it is converted to sorbitol. Under normal conditions, approximately 4% of the total glucose concentration in the lens is converted in this way [39].

As the concentration of glucose in the blood increases, the amount of it in the aqueous humor also increases, and consequently, the same happens in the lens. Hexokinase is then inhibited in the feedback mechanism by glycolysis products, while the sorbitol cycle becomes the main pathway involved in glucose metabolism. Aldose reductase (AR), which reduces glucose to sorbitol, is the key enzyme in this cycle [40]. The sorbitol dehydrogenase enzyme then catalyzes the conversion of sorbitol to fructose. Unfortunately, due to the low affinity of the latter enzyme, a large portion of the sorbitol is accumulated in the lens before further transformation. This property, combined with the poor permeability of the lens for sorbitol, results in its deposition in the lens [39].

The sorbitol pathway, also known as the polyol pathway, is the main pathway involved in diabetic cataract formation. An increased accumulation of sorbitol and a lower concentration of fructose in the lens result in a hyperosmotic effect, which results in the fluid moving into the interior of the lens to equalize the osmotic gradient. Initially, the energy-dependent lens pumps are capable of compensating for this but, eventually, become overloaded. This results in liquefaction and swelling of the lens fibers, leading to their degeneration and opacification [41,42]. This process is of particular relevance in young patients with type 1 diabetes [43,44] due to extensive edema of the cortical lens fibers [45].

Numerous publications suggest that the main enzyme of the polyol pathway—aldose reductase (AR)—plays a key role in the pathogenesis of diabetic cataracts. Snow et al. demonstrated that an increase in the AR gene expression may increase the risk of cataract development in diabetic patients [38]. Another study on the participation of AR in the pathogenesis of diabetic changes was conducted by Oishi et al., showing a positive correlation between the level of aldose reductase in red blood cells of patients under 60 years of age, with short-term diabetes, and developed posterior subcapsular cataract [46]. On the other hand, the correlation between the level of AR in erythrocytes and the density of lens epithelial cells was negative [47]. Of course, the density of lens epithelial cells in diabetic cataract patients is reduced, which is confirmed by, among others, Laspias et al. [48]. A research-based conclusion was made about the possibility of treating diabetic cataracts with AR inhibitors. Thus far, the effectiveness of such a procedure has been proven in animal studies (rats and dogs) [49]. Animal models such as the streptozotocin (STZ) rat, a model used to chemically induce type 1 diabetes by destroying pancreatic betacells, and galactose-fed animals, a model for obese type 2 diabetes, have been used to understand the mechanisms of diabetic cataract formation [50]. Unfortunately, it has not been confirmed in humans. The failure of aldose reductase inhibitors to slow the progression of cataracts in humans lies in the differences in AR activity and polyol accumulation between rats and humans, with humans exhibiting low levels of AR activity relative to rats. Moreover, the animal model mimicked the sudden onset of cataracts, as seen in patients with uncontrolled hyperglycemia. However, in most patients, cataracts develop after several years into the disease [50].

### 1.3. Oxidative Stress

Evidence of the involvement of oxidative stress in the development of diabetic cataracts is also available [51,52,53]. The induction of this process in diabetes results from a variety of mechanisms. Hyperglycemia may lead to auto-oxidation of glucose and non-enzymatic glycation, collectively referred to as glycoxidation, resulting in the overproduction of free oxygen radicals in the lens.

Non-enzymatic glycation of lens proteins leads to the formation of advanced glycation end products (AGEs). Glycation alters protein conformation and stability [54,55], induces protein aggregation and cross-linking, and increases susceptibility to proteolysis [55,56,57,58]. During the formation of AGEs, reactive oxygen species (ROS) are formed. Examples of these include superoxide anion radicals (O_2_^−^) and hydrogen peroxide (H_2_O_2_) [59,60]. Moreover, the binding of AGEs to a specific receptor for advanced glycation end-products (RAGEs) on the cell surface activating the NADPH oxidase (Nox) and NF-κB [61] becomes an additional signal for intracellular ROS formation [62]. Because the formation of AGEs cannot be reversed, even if hyperglycemia improves, their accumulation in the lens will continue to cause oxidative stress [63]. AGEs have been confirmed to occur both in the epithelium [64] and in the lens fibers [65,66]. Numerous studies indicate an elevated level of AGEs both in the lenses and in the serum of diabetic patients [67,68,69,70,71]. By binding with each other and with long-lived proteins, AGEs form a network of cross-links. All of the described processes concern proteins that constitute 33% of the total mass of the lens [39]. This is twice as much as in most other tissues [39]. Structural proteins called crystallins largely determine the transparency of the eye lens. Therefore, the molecular damage they undergo contributes to the formation of cataracts. Crystallins are water-soluble proteins categorized into two groups—α-crystallins and βγ-crystallins. α-crystallins are a class of small heat shock proteins. In addition to their structural role, α-crystallins have been shown to play a chaperone-like function by capturing unfolded or denatured aggregation-prone proteins and maintaining their refoldable conformation [72,73,74]. The protective function of α-crystallins relates to other crystallins, enzymes, and cytoskeletal proteins. The chaperoning ability of α-crystallin is believed to be essential for the maintenance of the transparency of the lens, thus, preventing the formation of cataracts [75]. There are numerous reports on the reduced protective activity of α-crystallins in the lenses of diabetic patients compared to the lenses of non-diabetic patients [76,77,78]. Therefore, many studies [75,77,79] have aimed to develop protective factors for the specific qualities of α-crystallins, with the goals of understanding the cause of cataracts and developing potential treatments.

Damage to antioxidant enzymes, such as superoxide dismutase or catalase, by protein glycation, among others, is responsible for the additional intensification of oxidative stress. The most dominant superoxide dismutase isoenzyme is copper–zinc superoxide dismutase 1 (SOD1), which degrades superoxide anions (O_2_^−^) to hydrogen peroxide (H_2_O_2_) and oxygen [80]. The importance of SOD1 in protecting against cataract development in the presence of diabetes has been demonstrated in various in vitro and in vivo animal studies [81,82,83]. Olofsson et al. observed that diabetic SOD1-null mice developed more cataracts than diabetic wild-type mice or non-diabetic mice [82,83]. Behndiga et al. noted similar results and, what is more, showed that the basal concentration of superoxide radicals is doubled in lenses lacking SOD1 [81]. 

The formation of highly reactive free radicals leads to DNA damage, protein oxidation, and lipid peroxidation, resulting in an increase in the amount of water-insoluble proteins [39]. It is well established that in a healthy lens, water-soluble proteins predominate, constituting up to 80% of all proteins. The fraction of water-insoluble proteins increases with age, which is a physiological process. However, in diabetes, it may proceed more rapidly and, as a result, lead to reduced lens translucency. The polyol pathway is also a mediator of oxidative stress [84]. The mechanism which contributes to oxidative stress is the usage of nicotinamide adenine dinucleotide phosphate (NADPH) during the conversion of glucose into sorbitol with the participation of AR. NADPH depletion impairs antioxidant defense because it is necessary for the formation of glutathione (GSH) from glutathione disulfide (GSSG) [85], and GSH acts indirectly as the primary scavenger of free radicals in the lens [39].

In addition, oxidative stress is generated during the conversion of sorbitol to fructose by sorbitol dehydrogenase (SDH) (i.e., the second step of the polyol pathway). At this stage, the nicotinamide adenine dinucleotide (NAD^+^) cofactor is reduced to NADH by SDH. NADH is a substrate for NADH oxidase, leading to the production of superoxide anions [86].

The polyol pathway of glucose metabolism and its relationship to oxidative stress is shown in Figure 2.

Excessive levels of glucose may disrupt the electron transport chain in the mitochondria. Therefore, the mitochondria found in the epithelial cells of the lens are another source of ROS in the lens. Increased expression of pro-inflammatory cytokines was also demonstrated in the lens epithelial cells of diabetic patients compared to the cells of non-diabetic patients [87]. This contributes to the dysfunction of these cells, consequently leading to their apoptosis, which has implications for the process of cataractogenesis [88,89].

In their study using a human lens epithelial cell line (HLEB3), Du S. et al. found that high glucose levels can directly induce oxidative stress and cell apoptosis. They also studied the effect of decorin on the p22^phox^–p38 pathway of HLEB3 cells under the conditions of high glucose levels. p22^phox^ and p38 have been described as important signaling factors responsible for the production of ROS under oxidative stress conditions in the mitochondria and have shown significantly increased expression of these proteins in the anterior lens capsule of patients with diabetic cataracts compared to patients with age-related cataracts. Their research suggests that decorin has a potential therapeutic role in the treatment of diabetic cataracts [90].

Currently, numerous studies confirming the inhibitory effect of antioxidants on the formation of diabetic cataracts in animals are being published. These are both exogenous antioxidants, such as alpha-lipoic acid [91] and vitamin E [92], and endogenous antioxidants, such as pyruvate [93]. Unfortunately, clinical observations in humans suggest limited effects of antioxidant compounds on the development of cataracts and indicate they might not be clinically significant [94].

### 1.4. Autophagy Lysosomal Pathway

Recent studies on the autophagy lysosomal pathway indicate its great importance in the formation of cataracts in hyperglycemia. This pathway is known to play an essential role in lens embryogenesis, being responsible for the autophagy of the nucleus and cell organelles during the differentiation of lens epithelial cells (LECs) into lens fiber cells (LFCs), ultimately forming the organelle-free zone. This pathway is also crucial in the process of removing damaged organelles and misfolded proteins, which are absorbed by autophagosomes and then degraded by lysosomal enzymes [95]. If these macromolecules cannot be digested by lysosomes, cells overloaded with autophagic substrates cause tissue malfunction. Researchers initially observed that autophagy was impaired in the LECs following exposure to hyperglycemic conditions [96].

Sun Y. et al. showed, for the first time, that hyperglycemia interferes with normal lysosomal degradation mediated by means of the transcription factor EB (TFEB) in a ROS-dependent manner, leading to autophagy blockage and, ultimately, lens opacity [97]. TFEB is a major lysosomal biogenesis gene playing a key role in maintaining LEC cellular homeostasis. The fact that the activation of TFEB by the curcumin analog, C1, increases lysosomal function, promotes autophagic flux, and eliminates oxidative molecules, thus, restoring the epithelial properties of the LEC and the transparency of the lens in conditions of elevated glucose levels [98], constitutes a valuable practical finding.

Studies have shown that the epithelial–mesenchymal transition (EMT), which is responsible for LECs losing their original polarity and tight junctions, occurs in a mouse model of diabetic cataracts (ENG number 63), and high glucose levels promote the EMT LEC processing both in vivo and in vitro [37,99]. EMT is the transdifferentiation of epithelial cells into mesenchymal cells with the activation of mesenchymal biomarkers such as α-smooth muscle actin (α-SMA) and the inhibition of epithelial cell biomarkers such as E-cadherin [100,101]. Abnormalities in the EMT process lead to the formation of LEC with the phenotypic features of mesenchymal cells with increased migration capacity, resistance to apoptosis, and the production of a greater amount of the extracellular matrix component (EMC), which also contributes to the formation of cataracts [102,103,104].

The autophagy and EMT processes may interact with each other through complex reactions [105]. It has been reported that they have common molecular mediators such as transforming growth factor beta (TGF-β), signal transducer and activator of transcription 3 (STAT3), integrins, and focal adhesion kinases (FAKs) [106]. The selective degradation of specific EMT proteins, such as Snail, appears to be the main molecular mechanism by which autophagy controls EMT [96]. Normally, Snail degrades rapidly, and its expression remains low. Similarly, p62 protein levels in cells are also very low as it is degraded in selective autophagy. However, autophagy is inhibited in a high-glucose environment, resulting in the accumulation of p62 in HLEB3. p62 may interact with Snail and protect it from degradation and, thus, increase its levels in cells. Snail moves from the cytoplasm to the nucleus and promotes the EMT process.

Li J. et al. demonstrated these complex relationships by reversing the processes caused by elevated glucose levels by the use of rapamycin. As a result, the autophagy-EMT pathway becomes a potential point of inhibition of diabetic cataracts [96].

### 1.5. Changes in the Metabolic Compositions in the Aqueous Humor

The aqueous humor (AH) not only supplies nutrients but also carries away metabolic wastes from avascular tissues in the eye. So far, little research has been conducted on the components of AH. Several studies have been performed to determine the differential profiles of lipid species in the AH between normal subjects and glaucoma patients [107,108]. Apart from glaucoma, distinct lipid profiles in AH have also been found in patients with conditions such as Fuchs endothelial corneal dystrophy [109] and polypoidal choroidal vasculopathy [110]. Promising conclusions were drawn by Wang et al. after having examined the differences in the lipid profile of the AH of patients with diabetic cataracts compared to the control group, which consisted of patients with normal glucose levels [111]. A multivariate analysis of lipid types showed significantly elevated concentrations of triglycerides (TGs) and diacylglycerol (DG), while ceramide-1-phosphates were significantly lower in AH in patients with diabetic cataracts. Moreover, TG concentration in AH was positively correlated with serum TG concentration in diabetic patients. On the basis of the conducted research, they hypothesized that AH lipid changes may contribute to the formation of cataracts, and lipid-targeted therapy may be a potential therapeutic strategy in diabetic cataracts [111].

An interesting metabolomic study was conducted by Pietrowska et al. in which they analyzed differences in AH small molecule composition between diabetic and non-diabetic patients with cataracts [112]. In the study, several molecules (methyltetrahydrofolic acid, taurine, niacinamide, xanthine, and uric acid) known for their antioxidant capabilities were found decreased in AH of diabetics. Another large group of metabolites significantly different between AH samples obtained from studied patients are amino acids (AA) [113] and their derivatives. The majority of these metabolites were significantly lower in AH of diabetics in comparison to non-diabetics. In general, AA may protect from cataracts or delay their development due to their antiglycating properties [114]. Differences that have been noticed confirm that increased oxidative stress and perturbations in amino acid metabolism in AH may be responsible for earlier cataract onset in diabetic patients. Additionally, the researchers paid special attention to tryptophan-related metabolites belonging to the AA group, which, except for their antiglycating properties, are efficient physical filters for the ultraviolet range. 

Assuming that tryptophan degradation products play an important role in the formation of senile cataracts, Andrzejewska-Buczko et al. examined the involvement of these compounds in the formation of diabetic cataracts. Their study found that in individuals with diabetes, the levels of kynurenine, 3-hydroxykynurenine, and anthranilic acid in the aqueous humor were higher compared to those without diabetes, while the levels of tryptophan and kynurenine were found to be similar in both groups. An accumulation of tryptophan and all its designated metabolites has been observed in the lenses of diabetic patients. The findings of this study also suggest that tryptophan degradation products may play an important role in the formation of diabetic cataracts [115].

### 1.6. The Influence of Diet on the Development of Cataracts

The complexity of diabetic cataract pathogenesis was taken into account by Cheng X. et al. in their study on the molecular mechanism of Protocatechualdehyde (PCA) in the treatment of diabetic cataracts [116]. PCA is a phenolic compound of plant origin with antioxidant and antiproliferative properties that may inhibit protein glycosylation [117,118]. It can also prevent the expression of AGE and TGF-β1 receptors in human lens epithelial cells cultured under diabetic conditions [119]. Based on the Network Pharmacology Study method, the study evaluated the multidirectional impact of PCA on the treatment of diabetic cataracts. The study has found that certain molecules—AKT1, MAPK3, and HDAC3—have been identified as potential targets for PCA, and they are associated with molecular functions such as protein kinase activity, protein kinase binding, and serine/threonine protein kinase activity. AKT1 plays an important role in the regulation of cell survival and can be activated by PI3K in diabetic cataracts [120]. MAPK, mitogen-activated protein kinase [121], HDAC3 (histone deacetylase 3) is a type of protease that plays an important role in the structural modification of chromosomes and the regulation of gene expression. HDAC is activated in age-related cataracts, and cataract formation is attenuated by HDAC inhibition [122,123,124]. It is possible that the concentration of HDAC in the eye increases in the presence of diabetic ocular complications [125]. Furthermore, the signaling pathways involved in these targets have been shown to mainly include the MAPK signaling pathway, the PI3K-Akt signaling pathway, and the AGE-RAGE signaling pathway in diabetic complications [116]. In the last stage of the study, an in vitro verification experiment was performed to improve the reliability of the results. AKT1, MAPK3, and HDAC3 transcription levels in glucose-induced human lens epithelial cells were significantly increased, and PCA treatment significantly inhibited AKT1, MAPK3, and HDAC3 mRNA levels in glucose-induced cells. In conclusion, the study’s results indicate that PCA may show a therapeutic effect on diabetic cataracts through various targets and signaling pathways. Additionally, it highlights the intricate nature of the pathogenesis of diabetic cataracts.

Resveratrol (RSV), belonging to the stilbene family, is another compound that demonstrates multidirectional possibilities of action. It exhibits a wide range of biological characteristics, including anti-glycation, antioxidant, anti-inflammatory, neuroprotective, and anticancer properties [126,127]. Resveratrol has been proven to inhibit the formation of selenite-induced cataracts due to the stimulation of increased levels of reduced glutathione (GSH) and by reducing the levels of malondialdehyde (MDA), which is a lipid peroxidation marker in rat lenses [128]. A study conducted on human lens epithelial cells has shown that RSV reduces cell death and ROS accumulation following an acute state of H_2_O_2_ oxidative stress. This protective effect in cells is due to the increased expression of enzymes such as superoxide dismutase-1 (SOD-1), catalase, and heme oxygenase-1 (HO-1) [129]. RSV has been shown to be effective in preventing and treating diabetic cataracts, as well as suppressing high-glucose-induced oxidative damage by activating p38 MAPK signaling to promote autophagy in HLECs [130]. These results suggest that RSV can be utilized in the prevention and treatment of diabetic cataracts [130]. Posterior capsule opacification has been shown to be possibly inhibited by RSV after cataract surgery [20]. It is worth noting that RSV supplementation is also effective in inhibiting the progression of diabetic retinopathy, including proliferative retinopathy [34].

## 2. Challenges and Future Directions

The increasing incidence of diabetes and an aging society require further research into the pathogenesis and potential targets of cataract therapy. Recent reports on the involvement of pyroptosis in eye diseases seem promising, including cataracts extensively discussed by Chen et al. [131], describing pyroptosis as an inflammatory form of cell death executed by gasdermins, a family of transmembrane pore-forming proteins activated via inflammasome-dependent or inflammasome-independent pathways. During the exploration of different types of cell death, they found that the relationship between cell death forms is complicated, and a myriad of ties between pyroptosis and other cell death in ocular diseases remain to be discovered. In addition, gene therapy and nanodrugs (i.e., drugs encapsulated with nanoparticles) targeting the pyroptosis pathway will also have satisfactory application prospects and are some of the research directions that are expected to achieve clinical breakthroughs in the future.

In recent years, an interesting subject is artificial intelligence (AI), which has had a profound and increasing impact on ophthalmology. AI has the potential to be a useful adjunctive tool for cataract management. First, AI can be applied as a telediagnostic platform to screen and diagnose patients with cataracts using slit-lamp and fundus photographs. The key challenges include ethical management of data, ensuring data security and privacy, demonstrating clinically acceptable performance, and improving the trust of end-users [113].

## 3. Conclusions

In summary, hyperglycemia induces oxidative stress, generating the formation of reactive oxygen species, increasing cell metabolism, intensifying the conversions in the polyol pathway, and the formation of advanced glycation end products (AGEs). Activation of all these processes is closely linked and is based on the principle of feedback loops [132].

Understanding these mechanisms is also important in everyday ophthalmological practice. The number of patients suffering from diabetes is constantly rising, and thus, the number of patients referred for cataract surgery is increasing. Thus, the question arises if we can offer non-surgical methods of treatment. This is all the more important because cataract surgery in diabetic patients is associated with a greater risk of both intra and postoperative complications. The moment of qualifying a patient for cataract surgery is often one of the stages of treatment of diabetic eye disease, whereas lens translucency plays a key role in both the diagnosis and the treatment of diabetic eye complications, such as diabetic macular edema or diabetic retinopathy.

## Figures and Tables

**Figure 1 ijms-24-06317-f001:**
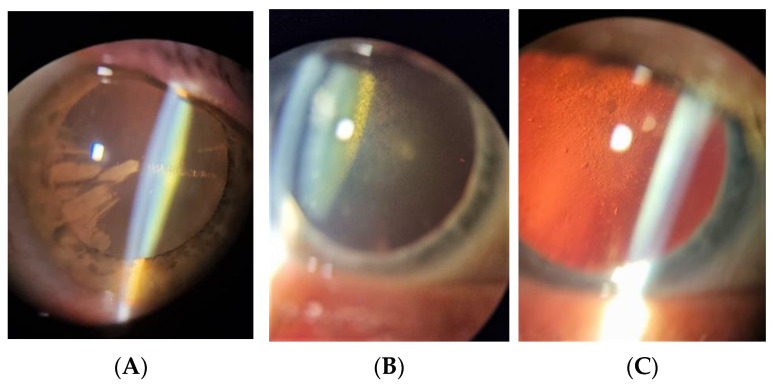
Cataracts related to diabetes. (**A**) Cortical cataract. (**B**) Posterior subcapsular cataract. (**C**) Posterior subcapsular cataract in retroillumination.

**Figure 2 ijms-24-06317-f002:**
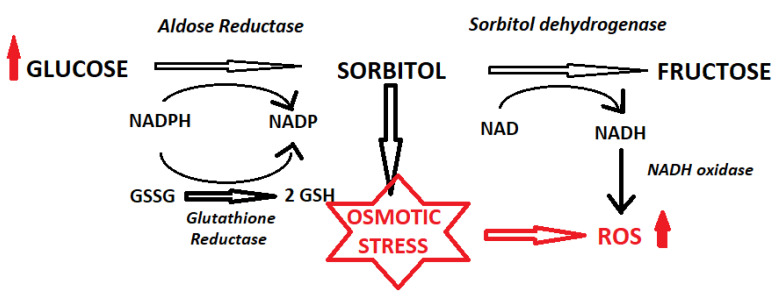
Polyol pathway of glucose metabolism and its effect on ROS.

## Data Availability

Not applicable.

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
