# Peer review of "Current Approach to the Pathogenesis of Diabetic Cataracts"

_ijms, 2023, doi:10.3390/ijms24076317_

Round 1
Reviewer 1 Report
General comments:
The authors describe and review the literature pathways contributing to diabetes-associated cataracts. It is well-written and and enjoyable read. The timeliness and expected citations of this article is uncertain. It does require more depth on the exact mechanism, even though not well known, that underlie cataract formation, and how all the pathways listed by the authors contribute to the current literature.
Major changes:
Page 3. 1.1 Diabetic cataract pathogenesis. Even if it is not fully understood, the composition of cataracts has been analyzed extensively. PMID: 31525139 PMID: 33849365 and many more. Please briefly explain the foundations of cataract formation, and eventually indicate how each of these pathways, in the context of diabetes, interact with these foundations.
Minor changes:
Page 2. Figure 1. “Shows a cataract” and not “shows cataract.
Author Response
Response to Reviewer 1 Comments
Thank you very much for your comments. Your valuable suggestions enriched the article in many paragraphs. I hope that the changes introduced will be in line with expectations.
Point 1: Page 3. 1.1 Diabetic cataract pathogenesis. Even if it is not fully understood, the composition of cataracts has been analyzed extensively.
Response 1: The paragraph has been rewritten.
A cataract is a form of opacification that develops in the translucent lens of the eye, ranging from partial to complete. This condition reduces the translucency of the lenses which results in reduced visual acuity for the patient, while making it difficult for the ophthalmologist to observe the posterior section of the eye and even making treatment impossible, as in the case of photocoagulation of the retina. The results of the clinical researches have shown that cataract development occurs more frequently and at an earlier age in diabetic compared to nondiabetic patients [2,3]. This fact is confirmed by the study of Memon et al. according to which under 40 years, 33.3% diabetics as compared to 16.4% non-diabetics develop cataract; the percentage was 41% and 14.5%, respectively, in 40-59 years age group. The same trend follows in the age group > 60 years where 16.4% non-diabetics develop cataract compared to 47% diabetics [3]. Similar results were obtained by Becker et al. in retrospective observational study involving 56,510 diabetes patients [4]. The estimated incidence rates of cataract were 20.4 (95% CI 19.8–20.9) per 1000 person-years (py) in patients with diabetes and 10.8 (95% CI 10.5–11.2) per 1000 py in the general population.
There are three types of cataract: cortical (CC), nuclear (NS), and posterior subcapsular (PSC). Each may occur independently or, more commonly, they co-occur with each other. Many studies have examined the likelihood of the occurrence of different types of cataracts in diabetic patients. Olafsdottir et al. showed that the prevalence of significant cortical, posterior subcapsular and nuclear cataract was 65.5%, 42.5% and 48.0%, respectively, in the type 2 diabetes population [5]. These results are consistent with most of the previous studies [6,7,8,9,10]. Two studies reported contrasting finding that is, NS was identified as the predominant subtype [3,11]. Cortical cataracts occurrence is often independent of sugar level fluctuations (Figure 1A) [12,5] while posterior subcapsular cataract is usually associated with poor glycemic control (Figure 1B,C) [5,13].
The association of T1DM and cataract is well documented in literature [14], the risk factors being the duration of diabetic symptoms before diagnosis, diabetic ketoacidosis, poor metabolic control, elevated HbA1c, possibly genetic factors, and treatment with glucocorticoids. Cataract affects from 0,7% to 3,4% of type 1 patients from different populations, and the average 10-year cumulative incidence of cataract surgery according to a study is 8.5% [15,16,17]. Regarding morphology of early diabetic cataract, Wilson et al. reported multiple morphologies including posterior subcapsular, lamellar, cortical, snowflake, and milky white type of early diabetic cataract [17], while most of other authors discriminate fewer types with posterior subcapsular cataract described as the most common type of diabetic cataract in childhood [18,19].
Point 2: Page 2. Figure 1. “Shows a cataract” and not “shows cataract.
Response 2: Description of the Figure 1 has been changed.
Figure 1 shows a cataract related to diabetes. A. Cortical cataract. B. Posterior subcapsular cataract C. Posterior subcapsular cataract in retroillumination.
Reviewer 2 Report
The authors presented a review about the diabetic cataract with a focus on its pathogenesis. Due to the increasing prevalence of diabetes worldwide, associated complications to the disease like the diabetic cataract are an increasing health issue. Thus, the subject is of high importance.
Unfortunately, the manuscript stays “superficial”, misses recent and key references and thus, cannot come to reasonable conclusions. For this reason, the article is recommended, regrettably, to be rejected in its present form.
Minor comments
1. Please remove the references from the abstract.
2. Please write molecular formulas correctly with superscripted/subscripted numbers (e.g., p. 4: O2- and H2O2).
3. In “Changes in the lipid profile in the aqueous humor”, p. 6, first sentence, it should be “after” instead of “After”. Please correct this.
4. Please uniform the format of the references to the main text.
Major comments
General
1. The general language is kind of “superficial”, sometimes not appropriate for a scientific journal (e.g., “clouding” of the lens) and not well written (e.g., p. 4 “...with the goal of understanding both the cause of cataracts and potential treatments”. Better would be: “…with the goals of understanding the cause of cataracts and to develop potential treatments.”).
2. The English needs to be reviewed.
3. Multiple of cited references are more than 10 sometimes 30 years old and need to be replaced by more recent articles. Consequently, parts of the review are not discussing recent knowledge and have to be rewritten.
4. Other phrases are missing a reference, e.g. “Even uncomplicated cataract surgery leads to an increase in the concentration of inflammatory cytokines, …VEGF, …IL-1, and PEDF” (Introduction, p. 2) or “On the other hand, the correlation between the level of AR in erythrocytes and the density of lens epithelial cells was negative” (1.2. the sorbitol pathway, p. 3). Please add appropriate references.
5. In my opinion the review misses a section about current and potential future treatment options, though a few are mentioned within the other sections.
Abstract
6. The first sentence is not correct and the reference is outdated: together with uncorrected refractive errors, cataract is the leading cause for reversible blindness worldwide, but mainly an issue in non-developed countries. In developed countries, cataract is not a big problem; here diabetic retinopathy (working age population) and AMD (elderly) are the bigger challenges. Newer information can be found in Steinmetz et al., DOI: 10.1016/S2214-109X(20)30489-7. Reference 2 is also more than 10 years old and should be replaced by a more recent one and the respective sentence modified if necessary.
Introduction
7. The sentence “Diabetic patients develop age-related changes in the lens that are indistinguishable from age-related cataracts in nondiabetic patients” is misleading. In young diabetic patients the changes are clearly not “age-related”. The phrase should be rewritten like, e.g., “…changes in the lens of diabetic patients resemble age-related modifications in elderly patients”.
8. The description “A cataract is a form of clouding that develops in the translucent lens of the eye, ranging from partial to complete” is not scientific and not specific enough. Please edit.
9. The description about what is a cataract, which forms exist, how can they be distinguished, which forms occur in what type of diabetes and if they are associated with poor glucose control or similar, is not detailed and scientific enough. It is not clear what is the prevalence of the different forms and if they occur in both types of diabetes, only type 1 or only type 2. Please rewrite the paragraph.
Diabetic cataract pathogenesis
10. First, it is written “….the lens is not vascularized…”. Later, the authors wrote “…changes in the vascularized tissues of the lens…”. This is a paradox that has to be clarified and corrected.
11. Overall, the section does not explain (or offers an hypothesis) the pathogenesis of the diabetic cataract. It is said i) that the pathogenesis is not fully understood, ii) that the mechanisms are different from other diabetic complications, iii) the role of hyperglycemia is important but hypertension and lipid dysregulation are not. However, what is suggested? What are the hypotheses? Please detail and answer these questions.
The sorbitol pathway
12. In the 4th paragraph it is written that AR shows a positive correlation in erythrocytes, short-term diabetes and PSC, but that – on the other hand – the correlation between AR levels in erythrocytes and the density of lens epithelial cells was negative. Then, the conclusion is made: “… a research-based conclusion was made about the possibility of treating diabetic cataracts with AR inhibitors. This is inconclusive, since the aforementioned results were contradictive. Please rewrite.
13. The considerations “It is worth noting that AR activity in humans is lower than in the animal model. Moreover, the animal model mimicked the sudden onset of cataracts as seen in patients with uncontrolled hyperglycemia. However, in most patients, cataract develops after several years into the disease [35].” Are not sufficiently discussed. It should be thoroughly explained that AR inhibitors might less efficient in human patients than in animals due to a lower enzymatic activity (in which animals, which organs, which conditions etc. has to be discussed). Additionally, the animal model has to be explained in more detail (sudden onset cataract) and which kind of diabetic cataract this model mimics (type of diabetes, prevalence etc.).
Oxidative stress
14. In the second paragraph, p. 4, it is written “….reduced protective activity of a-crystallins in the lenses of diabetic patients compared to….patients with normal glucose levels…”. This is not specific enough. Does it mean diabetic patients with badly controlled blood sugar? Or does it mean diabetic patients in general, independent from their glucose levels (then the second part of the phrase is not correct/complete)? Please rewrite.
15. On page 4 it is mentioned that SOD1 protects against cataract in “various in vitro and in vivo animal studies”. Please explain the protective mechanisms and the cited studies.
Changes in the lipid profile in the aqueous humor”
16. P. 6, first sentence: the reference is missing.
17. The section is unbalanced short compared to “Oxidative stress” and others. Does it refer only to 1 publication? If yes, this has to be extended. The same applies for the sections “Tryptophan degradation products in the aqueous humor” and “Genetics”.
The influence of diet on the development of cataracts
18. The phrase on p. 7, “Furthermore, the signaling pathways involved in these targets have been shown to mainly include the MAPK signaling pathway, the PI3K-Akt signaling pathway, and the AGE-RAGE signaling pathway in di-abetic complications” misses a reference.
Conclusion
19. The conclusion implicates that hyperglycemia that leads to AGE formation might be responsible for diabetic cataract. However, today, many patients control their glucose level pretty well and show nevertheless diabetic complications as diabetic cataract and others. Thus, the conclusion is not correct or at least, not well explained. Please rewrite.
Author Response
Response to Reviewer 2 Comments
Thank you very much for your comments. Your valuable suggestions enriched the article in many paragraphs. I hope that the changes introduced will be in line with expectations.
Point 1: The general language is kind of “superficial”, sometimes not appropriate for a scientific journal
Response 1: language has been edited.
Point 2: The English needs to be reviewed.
Response 2: The English has beed reviewed.
Point 3: Multiple of cited references are more than 10 sometimes 30 years old and need to be replaced by more recent articles.
Response 3: The references have been rewritten.
Point 4: Other phrases are missing a reference,
Response 4: The references have been added.
Even uncomplicated cataract surgery leads to an increase in the concentration of inflammatory cytokines, vascular endothelial growth factor (VEGF), hepatocyte growth factor, interleukin-1 (IL-1), and pigment epithelium-derived factor [33].
[33]. Dong N, Xu B, Wang B, Chu L, Tang X. Aqueous cytokines as predictors of macular edema in patients with diabetes following uncomplicated phacoemulsification cataract surgery. Biomed Res Int. 2015;2015:126984.
On the other hand, the correlation between the level of AR in erythrocytes and the density of lens epithelial cells was negative [47].
[47] Kumamoto Y, Takamura Y, Kubo E, Tsuzuki S, Akagi Y. Epithelial cell density in cataractous lenses of patients with diabetes: association with erythrocyte aldose reductase. Exp Eye Res. 2007 Sep;85(3):393-9.
Point 5: In my opinion the review misses a section about current and potential future treatment options, though a few are mentioned within the other sections.
Response 5: The ‘Challenges and future directions’ section has been added
Challenges and future directions
The increasing incidence of diabetes and an aging society require further research into the pathogenesis and potential targets of cataract therapy. Recent reports on the involvement of pyroptosis in eye diseases seem promising, including cataract extensively discussed by Chen et al. [130]. Descrobed pyroptosis as an inflammatory form of cell death executed by gasdermins, a family of transmembrane pore-forming proteins activated via inflammasome-dependent or inflammasome-independent pathways. During the exploration of different types of cell death, they found that the relationship between cell death forms is complicated, and myriad ties between pyroptosis and other cell death in ocular diseases remain to be discovered. In addition, gene therapy and nanodrugs (i.e., drugs encapsulated with nanoparticles) targeting the pyroptosis pathway will also have satisfactory application prospects and are some of the research directions that are expected to achieve clinical breakthroughs in the future.
In recent years, an interesting subject is artificial intelligence (AI), which has had a profound and increasing impact on ophthalmology. AI has the potential to be a useful adjunctive tool for cataract management. First, AI can be applied as a telediagnostic platform to screen and diagnose patients with cataract using slit-lamp and fundus photographs. The key challenges include ethical management of data, ensuring data security and privacy, demonstrating clinically acceptable performance, and improving the trust of end-users [113].
Point 6: The first sentence is not correct and the reference is outdated.
Response 6: The Abstract has been changed in line with your comments.
Abstract
Cataract remains the first or second leading cause of blindness in all world regions. In the diabetic population, cataract not only has a 3-5 times higher incidence than in the healthy population but also affects people at a younger age. In patients with type 1 diabetes, cataract occurs on average 20 years earlier than in the non-diabetic population. In addition, the risk of developing cataracts increases with the duration of diabetes and poor metabolic control. A better understanding of the mechanisms leading to the formation of diabetic cataracts enables more effective treatment and a holistic approach to the patient.
Point 7: The sentence “Diabetic patients develop age-related changes in the lens that are indistinguishable from age-related cataracts in nondiabetic patients” is misleading.
Response 7: The phrase has been edited
Changes in the lens of diabetic patients resemble age-related modifications in elderly patients.
Point 8: The description “A cataract is a form of clouding that develops in the translucent lens of the eye, ranging from partial to complete” is not scientific and not specific enough.
Response 8: The sentence on the definition of cataract has been changed.
A cataract is a form of opacification that develops in the translucent lens of the eye, ranging from partial to complete.
Point 9: The description about what is a cataract, which forms exist, how can they be distinguished, which forms occur in what type of diabetes and if they are associated with poor glucose control or similar, is not detailed and scientific enough. It is not clear what is the prevalence of the different forms and if they occur in both types of diabetes, only type 1 or only type 2.
Response 9: The paragraph has been rewritten.
A cataract is a form of opacification that develops in the translucent lens of the eye, ranging from partial to complete. This condition reduces the translucency of the lenses which results in reduced visual acuity for the patient, while making it difficult for the ophthalmologist to observe the posterior section of the eye and even making treatment impossible, as in the case of photocoagulation of the retina. The results of the clinical researches have shown that cataract development occurs more frequently and at an earlier age in diabetic compared to nondiabetic patients [2,3]. This fact is confirmed by the study of Memon et al. according to which under 40 years, 33.3% diabetics as compared to 16.4% non-diabetics develop cataract; the percentage was 41% and 14.5%, respectively, in 40-59 years age group. The same trend follows in the age group > 60 years where 16.4% non-diabetics develop cataract compared to 47% diabetics [3]. Similar results were obtained by Becker et al. in retrospective observational study involving 56,510 diabetes patients [4]. The estimated incidence rates of cataract were 20.4 (95% CI 19.8–20.9) per 1000 person-years (py) in patients with diabetes and 10.8 (95% CI 10.5–11.2) per 1000 py in the general population.
There are three types of cataract: cortical (CC), nuclear (NS), and posterior subcapsular (PSC). Each may occur independently or, more commonly, they co-occur with each other. Many studies have examined the likelihood of the occurrence of different types of cataracts in diabetic patients. Olafsdottir et al. showed that the prevalence of significant cortical, posterior subcapsular and nuclear cataract was 65.5%, 42.5% and 48.0%, respectively, in the type 2 diabetes population [5]. These results are consistent with most of the previous studies [6,7,8,9,10]. Two studies reported contrasting finding that is, NS was identified as the predominant subtype [3,11]. Cortical cataracts occurrence is often independent of sugar level fluctuations (Figure 1A) [12,5] while posterior subcapsular cataract is usually associated with poor glycemic control (Figure 1B,C) [5,13].
The association of T1DM and cataract is well documented in literature [14], the risk factors being the duration of diabetic symptoms before diagnosis, diabetic ketoacidosis, poor metabolic control, elevated HbA1c, possibly genetic factors, and treatment with glucocorticoids. Cataract affects from 0,7% to 3,4% of type 1 patients from different populations, and the average 10-year cumulative incidence of cataract surgery according to a study is 8.5% [15,16,17]. Regarding morphology of early diabetic cataract, Wilson et al. reported multiple morphologies including posterior subcapsular, lamellar, cortical, snowflake, and milky white type of early diabetic cataract [17], while most of other authors discriminate fewer types with posterior subcapsular cataract described as the most common type of diabetic cataract in childhood [18,19].
Point 10: First, it is written “….the lens is not vascularized…”. Later, the authors wrote “…changes in the vascularized tissues of the lens…”. This is a paradox that has to be clarified and corrected.
Response 10: Obviously this statement was incorrect.
Whereas, factors such as hypertension or lipid metabolism disorders, which contribute to the formation of diabetic changes in the vascularized tissues of the eye, are not significant in the pathogenesis of cataracts.
Point 11: Overall, the section does not explain (or offers an hypothesis) the pathogenesis of the diabetic cataract.
Response 11: Hypothesis has been added.
In this review, we will develop the hypothesis that the diabetic cataract formation can be caused by multifactorial reactions and pathways interacting with each other.
Point 12: In the 4th paragraph it is written that AR shows a positive correlation in erythrocytes, short-term diabetes and PSC, but that – on the other hand – the correlation between AR levels in erythrocytes and the density of lens epithelial cells was negative. Then, the conclusion is made: “… a research-based conclusion was made about the possibility of treating diabetic cataracts with AR inhibitors. This is inconclusive, since the aforementioned results were contradictive.
Response 12: To explain why comparing the correlation between AR in erythrocytes and PSC with the correlation between AR and epithelial cells density is not contradictory the clarification has beed added.
Of course, the density of lens epithelial cells in diabetic cataract patients is reduced, which is confirmed by, among others, Laspias et al. [48].
[48] Laspias GA, Thomopoulou GH, Lazaris AC, Kavantzas N, Koutselini H, Pagonis N, Tsapeli E, Politi E. Cytomorphometric study of epithelial cells in normal and cataractous human lenses in relation with hyperglycemia. Int Ophthalmol. 2016 Apr;36(2):147-58.
Point 13: The considerations “It is worth noting that AR activity in humans is lower than in the animal model. Moreover, the animal model mimicked the sudden onset of cataracts as seen in patients with uncontrolled hyperglycemia. However, in most patients, cataract develops after several years into the disease [35].” Are not sufficiently discussed.
Response 13: The paragraph has been corrected
Numerous publications suggest that the main enzyme of the polyol pathway – aldose reductase (AR), plays a key role in the pathogenesis of diabetic cataracts. Snow et al. demonstrated that an increase in the AR gene expression may increase the risk of cataract development in diabetic patients [38]. Another study on the participation of AR in the pathogenesis of diabetic changes was conducted by Oishi et al., showing a positive correlation between the level of aldose reductase in red blood cells of patients under 60 years of age, with short-term diabetes, and developed posterior subcapsular cataract [46]. On the other hand, the correlation between the level of AR in erythrocytes and the density of lens epithelial cells was negative [47]. Of course, the density of lens epithelial cells in diabetic cataract patients is reduced, which is confirmed by, among others, Laspias et al. [48]. A research-based conclusion was made about the possibility of treating diabetic cataracts with AR inhibitors. Thus far, the effectiveness of such a procedure has been proven in animal studies (rats and dogs) [49]. Animal models such as the streptozotocin (STZ) rat, a model used to chemically induce Type 1 diabetes by destroying pancreatic betacells, and galactose fed animals, a model for obese Type 2 diabetes, have been used to understand the mechanisms of diabetic cataract formation [50]. Unfortunately, it has not been confirmed in humans. The failure of aldose reductase inhibitors to slow the progression of cataract in humans lies with the differences in AR activity and polyol accumulation between rats and humans, with humans exhibiting low levels of AR activity relative to rat. Moreover, the animal model mimicked the sudden onset of cataracts as seen in patients with uncontrolled hyperglycemia. However, in most patients, cataract develops after several years into the disease [50].
Point 14: In the second paragraph, p. 4, it is written “….reduced protective activity of a-crystallins in the lenses of diabetic patients compared to….patients with normal glucose levels…”. This is not specific enough.
Response 14: The sentence has been changed.
There are numerous reports on the reduced protective activity of α-crystallins in the lenses of diabetic patients compared to the lenses of non-diabetic patients.
Point 15: On page 4 it is mentioned that SOD1 protects against cataract in “various in vitro and in vivo animal studies”. Please explain the protective mechanisms and the cited studies.
Response 15: This section has beed expanded.
The most dominant superoxide dismutase isoenzyme is copper-zinc superoxide dismutase 1 (SOD1), which degrades superoxide anions (O2−) to hydrogen peroxide (H2O2) and oxygen [80]. The importance of SOD1 in protecting against cataract development in the presence of diabetes has been demonstrated in various in vitro and in vivo animal studies [81-83]. Olofsson et al. observed that the diabetic SOD1-null mice developed more cataract than did the diabetic wild-type mice or the nondiabetic mice [82,83]. Behndiga et al. noted similar results and what is more, showed that the basal concentration of superoxide radicals is doubled in lenses lacking the SOD1 [81].
Point 16: P. 6, first sentence: the reference is missing.
Response 16: The reference has been added.
Promising conclusions were drawn by Wang et al. after having examined the differences in the lipid profile of the aqueous humor (AH) of patients with diabetic cataracts compared to the control group, which consisted of patients with normal glucose levels [111].
- Wang J, Zhang Y, Li W, Zhou F, Li J. Changes in the Lipid Profile of Aqueous Humor From Diabetic Cataract Patients. Transl Vis Sci Technol. 2022 Nov 1;11(11):5.
Point 17: The section is unbalanced short compared to “Oxidative stress” and others. Does it refer only to 1 publication? If yes, this has to be extended. The same applies for the sections “Tryptophan degradation products in the aqueous humor”
Response 17: The section ‘Changes in the metabolic compositions in the aqueous humor’ has been expanded.
Changes in the metabolic compositions in the aqueous humor
Aqueous humor (AH) not only supplies nutrients but also carries away metabolic wastes from avascular tissues in the eye. So far, few research has been done on the components of AH. Several studies have been performed to determine the differential profiles of lipid species in the AH between normal subjects and glaucoma patients [107,108]. Apart from glaucoma, distinct lipid profiles in AH have also been found in patients with conditions such as Fuchs endothelial corneal dystrophy [109] and polypoidal choroidal vasculopathy [110]. Promising conclusions were drawn by Wang et al. after having examined the differences in the lipid profile of the AH of patients with diabetic cataracts compared to the control group, which consisted of patients with normal glucose levels [111]. A multivariate analysis of lipid types showed significantly elevated concentrations of triglycerides (TG) and diacylglycerol (DG), while ceramide-1-phosphates was significantly lower in AH in patients with diabetic cataracts. Moreover, TG concentration in AH was positively correlated with serum TG concentration in diabetic patients. Based on the conducted research, they hypothesized that AH lipid changes may contribute to the formation of cataracts, and lipid-targeted therapy may be a potential therapeutic strategy in diabetic cataracts [111].
An interesting metabolomic study was conducted by Pietrowska et al. in which they analyzed differences in AH small molecule composition between diabetic and non-diabetic patients with cataract [112]. In the study several molecules (methyltetrahydrofolic acid, taurine, niacinamide, xanthine, and uric acid) known from their antioxidant capabilities were found decreased in AH of diabetics. Another large group of metabolites significantly different between AH samples obtained from studied patients are amino acids (113) and their derivatives. The majority of these metabolites were significantly lower in AH of diabetics in comparison to non-diabetics. In general AA may protect from cataract or delay its development due to their antiglycating properties [114]. Differences that have been noticed confirm that increased oxidative stress and perturbations in amino acid metabolism in AH may be responsible for earlier cataract onset in diabetic patients. Additionally, the researchers paid special attention to tryptophan-related metabolites belonging to AA group, which except antiglycating properties, is efficient physical filter for ultraviolet range.
Assuming that tryptophan degradation products play an important role in the formation of senile cataracts, Andrzejewska-Buczko et al. examined the involvement of these compounds in the formation of diabetic cataracts. The study found that in individuals with diabetes, the levels of kynurenine, 3-hydroxykynurenine and anthranilic acid in the aqueous humor were higher compared to those without diabetes, while the levels of tryptophan and kynurenine were found to be similar in both groups. Accumulation of tryptophan and all its designated metabolites has been observed in lenses of diabetic patients. The findings of this study also suggest that tryptophan degradation products may play an important role in the formation of diabetic cataracts [115].
Point 18: The phrase on p. 7, “Furthermore, the signaling pathways involved in these targets have been shown to mainly include the MAPK signaling pathway, the PI3K-Akt signaling pathway, and the AGE-RAGE signaling pathway in di-abetic complications” misses a reference.
Response 18: The reference has been added.
Furthermore, the signaling pathways involved in these targets have been shown to mainly include the MAPK signaling pathway, the PI3K-Akt signaling pathway, and the AGE-RAGE signaling pathway in diabetic complications [116].
- Cheng X, Song Z, Wang X, Xu S, Dong L, Bai J, Li G, Zhang C. A Network Pharmacology Study on the Molecular Mechanism of Protocatechualdehyde in the Treatment of Diabetic Cataract. Drug Des Devel Ther. 2021 Sep 23;15:4011-4023.
Point 19: The conclusion implicates that hyperglycemia that leads to AGE formation might be responsible for diabetic cataract. However, today, many patients control their glucose level pretty well and show nevertheless diabetic complications as diabetic cataract and others. Thus, the conclusion is not correct or at least, not well explained.
Response 19: Hyperglycemia that leads to AGE formation might be responsible for diabetic cataract but it’s not the only risk factor. In my opinion the explanation is in Diabetic cataract pathogenesis paragraph.
As clinical practice demonstrates, diabetic complications do not develop in a certain percentage of patients, while in others, even with adequate metabolic control, the progression of complications is surprisingly fast. These observations indicate that there are several risk factors that may affect the likelihood of developing diabetic cataracts [38].
Reviewer 3 Report
Title
OK
Abstract
The statement “Cataract in diabetic patients is the leading cause of blindness in developed and developing countries” needs to be rephrased because cataract is the leading cause of blindness globally, but ‘cataract in diabetic patients …’ is not the leading cause. The authors can refer to WHO references on the causes of blindness
Introduction
Paragraph 3 – the statement “This condition reduces the translucency of the lenses which results in reduced visual acuity for the patient, while making it difficult for the ophthalmologist to observe the posterior section of the eye and even making treatment impossible, as in the case of photocoagulation of vision” – it is not photocoagulation of vision but rather ‘photocoagulation of the retina’ – please correct this.
Section 1.1 Diabetic Cataract Pathogenesis
Paragraph 1 – the statement “Whereas, factors such as hypertension or lipid metabolism disorders, which contribute to the formation of diabetic changes in the vascularized tissues of the lens, …” – did the authors mean to say ‘vascularized tissues of the eye’ which would be correct. What do they mean by vascularized tissues of the lens which is avascular?
Conclusions
This is an important section of the article – but seems very superficial. The authors may wish to revisit this and expand this section to highlight key areas and also provide a future outlook.
General Comment
Since this is a review article, it would be useful to add a small section before the Conclusions about areas for future research or new and emerging research taking place in this field. On reading the article, this area seemed missing.
Author Response
Response to Reviewer 3 Comments
Thank you very much for your comments. Your valuable suggestions enriched the article in many paragraphs. I hope that the changes introduced will be in line with expectations.
Point 1: Abstract
The statement “Cataract in diabetic patients is the leading cause of blindness in developed and developing countries” needs to be rephrased because cataract is the leading cause of blindness globally, but ‘cataract in diabetic patients …’ is not the leading cause.
Response 1: The abstract has been edited and the reference number 1 has been removed as recommended by the secon Reviewer.
Abstract
Cataract remains the first or second leading cause of blindness in all world regions. In the diabetic population, cataract not only has a 3-5 times higher incidence than in the healthy population but also affects people at a younger age. In patients with type 1 diabetes, cataract occurs on average 20 years earlier than in the non-diabetic population. In addition, the risk of developing cataracts increases with the duration of diabetes and poor metabolic control. A better understanding of the mechanisms leading to the formation of diabetic cataracts enables more effective treatment and a holistic approach to the patient.
Point 2: Paragraph 3 – the statement “This condition reduces the translucency of the lenses which results in reduced visual acuity for the patient, while making it difficult for the ophthalmologist to observe the posterior section of the eye and even making treatment impossible, as in the case of photocoagulation of vision” – it is not photocoagulation of vision but rather ‘photocoagulation of the retina’ – please correct this.
Response 2: Obviously this statement was incorrect.
This condition reduces the translucency of the lenses which results in reduced visual acuity for the patient, while making it difficult for the ophthalmologist to observe the posterior section of the eye and even making treatment impossible, as in the case of photocoagulation of the retina.
Point 3: Paragraph 1 – the statement “Whereas, factors such as hypertension or lipid metabolism disorders, which contribute to the formation of diabetic changes in the vascularized tissues of the lens, …” – did the authors mean to say ‘vascularized tissues of the eye’ which would be correct.
Response 3: Of course this statement has been changed.
Whereas, factors such as hypertension or lipid metabolism disorders, which contribute to the formation of diabetic changes in the vascularized tissues of the eye, are not significant in the pathogenesis of cataracts.
Point 4: Conclusions
This is an important section of the article – but seems very superficial. The authors may wish to revisit this and expand this section to highlight key areas and also provide a future outlook.
Point 5: General Comment
Since this is a review article, it would be useful to add a small section before the Conclusions about areas for future research or new and emerging research taking place in this field. On reading the article, this area seemed missing.
Response 4 and 5: The ‘Challenges and future directions’ section has been added
Challenges and future directions
The increasing incidence of diabetes and an aging society require further research into the pathogenesis and potential targets of cataract therapy. Recent reports on the involvement of pyroptosis in eye diseases seem promising, including cataract extensively discussed by Chen et al. [130]. Descrobed pyroptosis as an inflammatory form of cell death executed by gasdermins, a family of transmembrane pore-forming proteins activated via inflammasome-dependent or inflammasome-independent pathways. During the exploration of different types of cell death, they found that the relationship between cell death forms is complicated, and myriad ties between pyroptosis and other cell death in ocular diseases remain to be discovered. In addition, gene therapy and nanodrugs (i.e., drugs encapsulated with nanoparticles) targeting the pyroptosis pathway will also have satisfactory application prospects and are some of the research directions that are expected to achieve clinical breakthroughs in the future.
In recent years, an interesting subject is artificial intelligence (AI), which has had a profound and increasing impact on ophthalmology. AI has the potential to be a useful adjunctive tool for cataract management. First, AI can be applied as a telediagnostic platform to screen and diagnose patients with cataract using slit-lamp and fundus photographs. The key challenges include ethical management of data, ensuring data security and privacy, demonstrating clinically acceptable performance, and improving the trust of end-users [113].
Round 2
Reviewer 1 Report
GENERAL COMMENTS:
The introduction has been approved and is adequate for publication.
Reviewer 2 Report
The authors presented a significantly edited version of the review about the diabetic cataract with a focus on its pathogenesis. Most comments were detailed and satisfyingly answered. Thus, I am glad to recommend the new version for publication after only minor revision.
Minor comments
1. Please write molecular formulas correctly with superscripted/subscripted numbers (e.g., p. 4: O2- and H2O2).
2. Page 1, introduction, 2nd paragraph: please add references.
3. Page 2, introduction, 2nd, paragraph: “…0,7% to 3,4%...“ has to be corrected to “…0.7% to 3.4%...“.